Field and laboratory metabolism and thermoregulation in rhinoceros auklets

Umeyama Aika 1
Niizuma Yasuaki niizuma@meijo-u.ac.jp 1
Shirai Masaki 2
1 Laboratory of Environmental Zoology, Faculty of Agriculture, Meijo University , Nagoya , Aichi , Japan
2 Environmental Science Research Laboratory, Central Research Institute of Electric Power Industry , Abiko , Chiba , Japan
Welch Jr. Kenneth
Electronic publication date: 2021 May 18
Publication date: 2021
Volume: 9
Electronic Location ID: e11460
Received 2020 Sep 22; Accepted 2021 Apr 25
Copyright: 2021 Umeyama et al.
Copyright year: 2021
Copyright holder: Umeyama et al.
License: This is an open access article distributed under the terms of the Creative Commons Attribution License, which permits unrestricted use, distribution, reproduction and adaptation in any medium and for any purpose provided that it is properly attributed. For attribution, the original author(s), title, publication source (PeerJ) and either DOI or URL of the article must be cited.
License URL: https://creativecommons.org/licenses/by/4.0/

Keywords: Energy cost on water, Daily energy expenditure, Doubly labeled water, Resting metabolic rate, Resting seasurface, Thermoregulation, Seabird, Thermal conductivity, Heat loss, Seasurface temperature

Funding: Co-operation Research Program of the Wildlife Research Centre, Kyoto University This work was supported by the Co-operation Research Program of the Wildlife Research Centre, Kyoto University. There was no additional external funding received for this study. The funders had no role in study design, data collection and analysis, decision to publish, or preparation of the manuscript.

==============================
Seabirds spend most of their lives at sea, except when visiting their breeding sites. Since the thermal conductivity of water is 25 times higher than that of air, seabirds resting on water lose heat and expend a considerable amount of energy for thermoregulation. For example, rhinoceros auklet (Cerorhinca monocerata), a medium-sized (480620 g) alcid, spends most of its time floating on the sea. In order to estimate the cost of this behavior in terms of their daily energy expenditure (DEE), we studied rhinoceros auklets breeding on Teuri Island, Hokkaido Japan. We measured their resting metabolic rate (RMR) in air and on water by respirometry, and estimated their DEE by the doubly labeled water method. While RMR on water did not vary significantly between 10C and 15C, it was significantly higher at 5C. Air temperature (5.020.0C) had no effect on RMR. The DEE of free-ranging auklets averaged 1,005.5kJday1 (130.2, n=3). Our results indicate that RMRs are elevated for auklets resting on water, particularly below their lower critical temperature (LCT), compared with in air. Accordingly, spending time above their LCT on water at any time of year will provide enhanced benefits, particularly to seabirds such as rhinoceros auklets which rest a considerable amount of time on water.

Introduction

The ability of endothermic animals to thermoregulate may affect their life history traits, foraging behavior, and their distributions (Jenssen, Ekker & Bech, 1989; Humphreys, Wanless & Bryant, 2006; Lovvorn et al., 2009). Endothermic animals living in or on the sea expend considerable energy on thermoregulation even while resting on the surface because the thermal conductivity of water is 25 times great than that of air (Kaseloo & Lovvorn, 2005; Niizuma et al., 2007). Bio-logging techniques have shown that adult seabirds spend significant amounts of time resting on the sea during their chick-rearing period (Wilson, Weimerskirch & Lys, 1995; Garthe, Grmillet & Furness, 1999; Falk et al., 2000; Daunt et al., 2002; Tremblay et al., 2003; Kato, Watanuki & Naito, 2003). In addition to measurements of the high cost behaviors such as flying and diving (Elliott et al., 2013a), it is important to assess their energy expenditure during resting in order to more fully understand their energetics.

Seabirds may have physiological adaptations for floating on cold sea water, which include reducing their thermal conductance by means of a thick water-repellent plumage (Kooyman et al., 1976; Jenssen, Ekker & Bech, 1989; Dawson et al., 1999) and reducing heat flow to the periphery via vasoconstriction in the skin and appendages (Johansen & Bech, 1983; Folkow & Blix, 1987; Niizuma et al., 2007). The air trapped in the loose tangle of air pockets formed by the barbs and barbules of their plumulaceous inner vanes is the main component of plumage insulation for seabirds (Dawson et al., 1999). However, this insulation may constrain their rate of heat dissipation while flying between their nesting and foraging areas because birds produce excess heat during energy-intensive flapping flight (Elliott et al., 2013a; Guillemette et al., 2016; Nord & Nilsson, 2019). This may be especially significant in the temperate zone, where seabirds are less able to lose heat in the mild climate, yet endothermic animals must dissipate their metabolic heat to avoid reaching lethal body temperatures (Speakman & Krl, 2010; Nilsson, Molokwu & Olsson, 2016).

The Alcidae, a group of seabirds that breed from the temperate zone to the Arctic, have relatively dense plumage, high wing loading and continuous fast flapping flight, deliver food to their nestlings during the chick-rearing period, but spend significant amounts of time resting on the sea (Wilson, Weimerskirch & Lys, 1995; Gaston, Anthony & Jones, 1998; Garthe, Grmillet & Furness, 1999; Falk et al., 2000; Daunt et al., 2002; Tremblay et al., 2003; Kato, Watanuki & Naito, 2003). The rhinoceros auklet (Cerorhinca monocerata), a medium sized (480620 g) member of the Alcidae, breeds on offshore islands in areas of temperate waters in the northern Pacific and migrates southward to wintering areas (Gaston, Anthony & Jones, 1998). The auklets rearing chicks on Teuri Island, Hokkaido, Japan, spend 18% of their time on land at the colony, 14% flying, 13% in diving related behavior, but 55% floating on the sea (Kato, Watanuki & Naito, 2003). During their annual movement, they experience various water temperatures ranging from cold (46C) in early March, associated with their northward migration and early arrival on the breeding grounds, to mild (1114C) during the winter from October to late February in the southwestern Sea of Japan Sea (Takahashi et al., 2015). The sea surface temperature around the Teuri Island breeding colony increases during the breeding season from about 5C in early April to 15C in early July (Ito et al., 2009). Lower critical temperature (LCT) is typically higher on water than in air, and the rate of increase in metabolic rate as ambient temperature decreases below LCTis often steeper on water than in air (e.g.,Stahel & Nicol, 1982; Gabrielsen, Mehlum & Karlsen, 1988). Because auklets spend up to 55% of their time resting on the sea, it is important to determinine whether water temperature influences energy expenditure while resting. However, little is known about how much energy rhinoceros auklets require for thermoregulation while resting on water at various water temperatures.

In order to elucidate the energy cost of the time spent resting on the sea for pelagic seabirds, the resting metabolic rate (RMR) and daily energy expenditure (DEE) of the rhinoceros auklet were estimated quantitatively. RMR both in the air and on water, at various ambient temperatures, was measured using respirometrythe most common technique for measuring energy expenditure (Halsey, 2011). The DEE of rhinoceros auklets rearing chicks was estimated using the doubly labeled water (DLW) methoda common technique for estimating energy expenditure of free-living animals (Shaffer, 2010). These data were then used to assess their LCT in air and on water, and the energy cost of resting on water as a proportion of their DEE while rearing chicks.

Materials & Methods

Study area and species

This study was carried out at Teuri Island (4425N, 14152E), in the northern Sea of Japan, off northwest Hokkaido, from May to July 20152107. About 300,000 pairs of rhinoceros auklets breed on the island in the largest single breeding colony in the world (Watanuki & Ito, 2012).

To measure the RMRs of adult auklets in air and on water using respirometry, 43 auklets were captured, using landing nets, as they returned to their nests at night. Individual birds were used for only one measurement of RMR in air or on water. Birds were captured and experimented upon under license from the Ministry of the Environment, Government of Japan.

During the chick-rearing period in 2017 specifically, 16 rhinoceros auklets were caught by hand or landing net at the nesting colony or in their nest burrows, five for measurements of background and initial isotope enrichment, and 11 for measurements of DEE by means of the DLW method.

The procedures used in this study were approved by the Animal Experimental Committee of Meijo University (2015-A-E-5, 2016-A-E-10, 2017-A-E-2). The fieldwork was permitted by the Ministry of the Environment (21-26-0291 0292, 21-27-0367 0368 0369 0370 0371, 21-28-0344 035 036 037) and the Agency of Cultural Affairs (26-4-2188, 27-4-1928, 29-4-18).

Measurements of resting metabolic rate using respirometry in air and on water

Oxygen consumption rate (Vo2) was measured using an open-flow respirometry system composed of an acrylic metabolic chamber and an oxygen analyzer (Xentra 4100, Servomex Ltd, UK) as previously described in Shirai et al. (2015). For the measurement of RMR in air, a 20-L metabolic chamber (20cm long 25 cm high 40 cm wide) was submerged in a thermostatic water bath and maintained at 4.720.7C. For the measurement of RMR on water, a 72-L metabolic chamber (30cm long 60 cm high 40 cm wide) was filled with freshwater (to a depth of 30cm) maintained at 5.517.5C.

The wild-caught auklets were placed in darkened boxes (30cm נ30 cm נ25 cm), transported from the colony to the field station situated within 10 min of the capture site, then kept for at least one hour to minimize the effects of capture stress on their metabolic rates (Shirai et al., 2013). After one hour, they were weighed to the nearest 5 g, using a Pesola spring balance. They were then placed in the metabolic chamber for 12h over night to measure their RMR. After finishing the measurements, they were weighed again and released on the colony at night. We assumed a linear decrease in body mass to estimate the body mass value used for calculating the mass-specific metabolic rate.

During measurements, the chamber was kept dark by covering it with a blackout curtain. The chamber temperature (Tc,0.3C) and atmospheric pressure (Pa, 1.5 hPa) were recorded (using a TR-73U Thermo Recorder T&D Corp.), and water temperature was measured every minute (0.3C, using a TR-52i Thermo Recorder T&D Corp.). The rate of airflow (VE) through the chamber was controlled at 2.0 Lmin1 in air and 3.0 Lmin1 on water using a mass flow controller (2%, Type HM1171A, Tokyo Keiso). The effluent air from the chamber was dried and a fraction of the dry outlet air was directed into the oxygen analyzer. Absorption of oxygen into water in the chamber was less than 0.0015% per minute (Allers & Culik, 1997). The oxygen analyzer was calibrated using dry effluent air (20.946% oxygen) and pure stock nitrogen (0.000% oxygen) before beginning each experiment. The oxygen concentration of the effluent air (FEO2) was recorded every minute by computer.

Vo2 was calculated using formula 3A in Withers (1977) as follows, VO2=VEFIO2FEO211RQFIO2.

RQ was the respiratory quotient, assumed to be 0.8 based on Koteja (1996). FIO2 was an oxygen concentration of influent air of 20.946%. A conversion coefficient was used 20.1 kJL1 in calculating energy expenditure (Schmidt-Nielsen, 1997). All results are given at standard temperature, pressure, and dryness (STPD).

As previously described in Shirai et al. (2013), we estimated RMR to be the minimum value recorded over a 20 min interval during the 12 h measurements (Table S1).

Measurements of daily energy expenditure using the doubly labeled water method

We obtained estimates of DEE in rhinoceros auklets using the single-sample approach of the DLW method as previously described in Niizuma & Shirai (2015). The method allowed an estimation of initial isotope enrichment from a single blood sample and was a less invasive technique with lower impact on the behavior of study subjects (Schultner et al., 2010; Niizuma & Shirai, 2015). Recent validation studies have demonstrated that the precision of the DLW technique can be increased by using a longer sampling interval and/or by applying it to a species with a higher metabolic rate (Shirai et al., 2015; Kume et al., 2019). The DLW injectate used in our study contained 21.0 atom percent 18O, 10.5 atom percent 2H, and 0.9% NaCl.

Blood samples from five wild-caught auklets taken between 21:0022:00 were used to determine mean background and initial levels of the 2H and 18O isotopes. After capturing the birds, one mL of blood was collected from the brachial vein as a background sample; then the DLW was injected into the body cavity. After the DLW injection, the auklets were kept individually in plastic boxes for 90min; then further 1-mL blood samples were collected from each individual as initial samples. After sampling, they were weighed with a Pesola spring balance accurate to the nearest 10 g; then released at the nesting site.

Eleven individuals were caught in their nest burrows with their chicks to investigate their DEE. DLW was injected into the abdominal cavity of each bird. After being weighed, all individuals were banded with individually numbered metal bands and released back into their nests. Four of the injected individuals were recaptured in their nest burrows at night after they had returned from foraging trips. Immediately after recapture, a final 1-mL blood sample was collected, and each bird was re-weighed. These procedures were conducted at night (21:0023:00) to mitigate breeding disturbance (Sun et al., 2020), and required less effort for recapturing birds when compared with previous studies involving attaching bird-borne data-loggers (Kuroki et al., 2003; Kato, Watanuki & Naito, 2003; Matsumoto et al., 2008). Therefore, the recovery rate in this study was relatively lower than in previous ones.

We quantified the injectate by weighing the syringe (to the nearest 0.0001 g with an electronic balance in the field laboratory) before and after each injection following Speakman (1997). On average, birds were injected with 3.1326 g DLW ( 0.0783 s.d.). We heparinized and centrifuged (5 min, 6200 rpm) all blood samples. After centrifugation, we stored each serum sample at 25C in a 0.5 mL screw-topped plastic vial with an O-ring (Asahi Techno Glass Co.) until isotopic analysis.

We diluted the serum and injectate samples with distilled water measured with an electronic balance (Mettler-Toledo, Columbus, OH, USA) to the nearest 0.01 mg. We analyzed the 2H and 18O isotope concentrations of the serum, DLW injectate, and distilled water using isotope ratio mass spectrometry (IRMS; Hydra 20-20, Sercon, Crewe, UK; Shirai et al., 2012; Shirai et al., 2015). We used the water equilibration method (Horita et al., 1989) to analyze the serum, DLW injectate, and distilled water in duplicate. Water standards (Iso-Analytical, Crewe, UK) were used to establish calibration curves for normalizing the values. Each sample was analyzed in duplicate. All isotope enrichments were measured in per mille relative to the working standards and converted to an absolute ratio for 2H by using equation 14.4, and for 18O by using equation 14.9, from Speakman (1997). Absolute ratios were converted to ppm using equations from Speakman (1997): equation 14.8 for 2H, and equation 14.14 for 18O. All subsequent calculations in the DLW method were performed on the mean values of each sample analyzed in duplicate.

Calculation of CO2 production rates in the field

Ideally, background and initial isotope levels should be determined for each subject (Speakman & Racey, 1987). However, since this increases both the handling time and disturbance of the subject, the background and initial isotope abundances were determined for just five individuals. The background isotope level averaged 2002.04 ppm (range 1999.752005.16 ppm) for 18O and 159.64 ppm (range 156.22165.77 ppm) for 2H. We used these mean background levels to calculate the CO2 production rate (rCO2, mL day1).

We also estimated initial isotope enrichment based on the relationship of increments for isotope injection (Hinc or Oinc, ppm) and body mass (BM, g) and respective DLW injectate established for the birds as previously described in Niizuma & Shirai (2015). Hinc=1915.0+3.835BM+17661.0Hinj27.141BMHinj,

Hi=Hinc+Hb,

Oinc=3875.7+8.639BM+36186.2Oinj60.213BMOinj,

Oi=Oinc+Ob,

where Hinj and Oinj represent the respective DLW injectate (2H or 18O, mol), Hi and Oi represent the estimated initial isotope enrichments and Hb and Ob represent the background isotope enrichments (2H or 18O, ppm). The Hinc equation has an adjusted R2 of 0.942, while the Oinc equation has an adjusted R2 of 0.952.

Using the DLW injectates, the background and the estimated initial isotope enrichments, we calculated the isotope dilution spaces for 18O (No, mol) using the general equation: No=OinjOiOdObOi

where Od represents the isotope concentration (2H or 18O, ppm) in the DLW injectate. To convert the units of the isotope dilution spaces, we used a conversion factor of 18.002 g mol1 (Speakman, 1997).

The turnover rates for 2H and 18O (kd and ko, respectively, day1) were determined using the following general equations:

kd=lnHiHblnHfHbt

ko=lnOiOblnOfObt

where Hf and Of represent the respective isotope concentrations (2H or 18O, ppm) of the final samples and t represents the time interval between the injection and final samples days (Lifson & McClintock, 1966; Speakman, 1997).

As previously described in Shirai et al. (2012), we used Speakmans (1997) one-pool model for calculating rCO2 in this study as follows: rCO2=N2.078kokd0.0062kdN

where N = No. To convert units in mLCO2 day1 into energy equivalents, it was assumed that one mL of CO2 equals 25.11 J (Gessaman & Nagy, 1988).

Statistical analysis

All statistical analyses were performed in R 3.3.2 (R Core Team, 2018). Mass-specific metabolic rates of rhinoceros auklets resting in air and on water were tested for mean differences among air and water temperatures using one-way analysis of variance (ANOVA). When significant differences were observed among temperatures, the TukeyKramer multiple-comparison test was applied to determine which means were significantly different.

Results

Measurements of resting metabolic rate in air and on water

The RMR of rhinoceros auklets (555.6 g39.6 s.d., n=27) in air was not affected by air temperature (F3,23=0.893, P=0.460; Fig. 1A). The RMR in air averaged 0.02580.0033 kJ g1 h1 (n=27).

Figure 1 Resting metabolic rate (A) in air and (B) on water at different ambient temperatures in rhinoceros auklets.

The RMR of the auklets (565.648.7 g, n=16) on water was affected significantly by water temperature (F2,13=8.32, P=0.0047; Fig. 1B). While RMR on water did not vary significantly between 10C (0.03660.0045 kJ g1 h1, n=5) and 15C (0.03470.0036 kJ g1 h1, n=6) (t14 = 0.686, P=0.780), it was significantly higher at 5C (0.04600.0060 kJ kg1 h1, n=5) (5 vs 15C: t14 = 3.894, P=0.0049; 5 vs 10C: t14 = 3.071, P=0.0226). Auklet RMR on water at combined temperatures of 10C and 15C was 0.03560.0040 kJ g1 h1, n=11).

Daily energy expenditure of chick-rearing rhinoceros auklets

Four birds were recaptured after foraging trips following DLW injection. Three were recaptured after one-day trips (24.10.3h), but one was recaptured after a three-day (72.5h) trip and was found to have almost equal the final isotopic enrichment to the background abundance. Therefore, calculations of DEE were only possible for three individuals. The DEE of free-ranging auklets, which initially weighed 556.3 g ( 42.0, n=3), averaged 1005.5 kJday1 ( 130.2, n=3). The DEE/RMR ratio (based on RMR in air) was 2.9.

Discussion

Air temperature was not found to affect adult rhinoceros auklet RMR over the range of temperatures measured. However, when RMRs were measured for adult rhinoceros auklets on water, there was an effect of temperature, with an increase in RMR at the lowest temperature (Fig. 1). Our measurements of RMR in air are similar to the 0.0248 kJ g1 h1 for basal metabolic rate (BMR; Shirai et al., 2013) and the value estimated from the allometric equation for the Charadriiformes (0.0259 kJ g1 h1; BMR = 2.149 m0.804 kJ/day, where m is body mass (556 g); (Ellis & Gabrielsen, 2002). Nonetheless, we acknowledge that capture may cause a stress response whereby birds subsequently spend considerable time preening on the water leading to a low DEE (Schultner et al., 2010), especially for auklets, which are known to be particularly sensitive to disturbance (Sun et al., 2020) as shown by our low recapture rate. The DEE was equal to 112% and within the confidence interval (5771276 kJday1) of the predicted DEE that was calculated (using latitude = 44, body mass = 556 g, and breeding phase = Brood) from an allometric equation for seabirds (Dunn, White & Green, 2018). This suggests that the measured DEE of rhinoceros auklets is reasonable in comparison with previous seabird studies.

Resting metabolic rate in air and on water

We were unable to demonstrate the existence of an LCT in air for rhinoceros auklets in this study, but suspect it to be at least lower than 5C. The LCT in air of seabirds decreases with body mass and latitude. Although the LCT for adult rhinoceros auklets on Teuri Island was estimated to be 13.6C from equation 11.9 in Ellis & Gabrielsen (2002), our results suggest that it is lower than the estimation. The LCT of rhinoceros auklets in air is similar to that of other seabird species such as common murre (Uria aalge), thick-billed murre (U. lomvia), dovekie (Alle alle), black guillemot (Cepphus grylle) and black-legged kittiwake (Rissa tridactyla) that breed in arctic regions (Johnson & West, 1975; Gabrielsen, Mehlum & Karlsen, 1988; Gabrielsen et al., 1991), but not northern fulmar (Fulmarus glacialis) which has an LCT in air of 9.0C (Gabrielsen, Mehlum & Karlsen, 1988). Cassins auklet breeding on Triangle Island, British Columbia, Canada (N 50), had an LCT in air of 16C (Richman & Lovvorn, 2011) which is higher than that of the rhinoceros auklet. Despite breeding in the temperate zone, the rhinoceros auklets in this study had similar thermal properties at LCT in air to those breeding in the Arctic. Their insulation properties would constrain their heat dissipation rate during flapping flight between their nesting and foraging areas, especially in the temperate zone (Guillemette et al., 2016; Nord & Nilsson, 2019). Alcidae are noted to have an energy expenditure that is 31 times greater than BMR during flight, which is the highest known for any vertebrate (Elliott et al., 2013a). Since Teuri Island is at the southern limit of this species breeding area in the west Pacific, rhinoceros auklets with a lower LCT in air would have difficulty in dissipating heat while flying with food from their foraging area to their nesting site due to their high level of insulation in air (Schraft, Whelan & Elliott, 2019).

In contrast to their RMR in air, we estimated the LCT on water of rhinoceros auklets between 5C and 10C. Their LCT on water is lower than that for common murre, thick-billed murre and Cassins auklet (Croll & McLaren, 1993; Richman & Lovvorn, 2011). This result could have important implications for their ecology. The sea surface temperature around Teuri Island increases from about 5C in early April to 15C in early July during the auklet breeding season (Ito et al., 2009). After breeding, the auklets migrate to more southerly areas where, from October to late February, they experience water temperatures of 1114C, but for a short period from early March to April associated with their northward migration they experience sea surface temperatures of 46C (Takahashi et al., 2015). Therefore, they could rest on the sea at minimum energetic cost during most seasons due to their LCT on water being lower than the usual sea surface temperature. However, foraging auklets may remain longer on the sea after diving to digest their food (Elliott et al., 2014) and thus increase their metabolic rate for the obligatory component of the heat increment of feeding (Hawkins et al., 1997), which may be used for thermoregulation on water.

The Energetic cost of resting on the sea surface

The DEE/RMR ratio provides an estimate of how much birds must increase baseline costs to forage and thermoregulate in a particular environment and may be intrinsically set by physiological constraints (within four times RMR in air) (Drent & Daan, 1980). The value in this study is below the proposed energetic ceiling level and within the range among Alcidae (2.73.8 reviewed in Ellis & Gabrielsen, 2002).

Since the sea surface temperature around Teuri Island during the chick-rearing period was 813C (Ito et al., 2009), the auklets can be assumed to expend their energy within their TLC while resting on the water around the breeding site. For rhinoceros auklets, the energy cost of resting on water is likely to be dependent on the time spent on water per day (%). Rhinoceros auklets spend up to 55% of their time on water (Kato, Watanuki & Naito, 2003) because they only deliver food to their chick once a day at most (Watanuki, 1987; Takahashi et al., 1999). Common murres at Witless Bay, Newfoundland spend longer resting on water (57.5% of their time) (Cairns et al., 1990) than those at Hornya (24.9%) (Tremblay et al., 2003). When capelin (Mallotus villosus) are present, common murres at Witless Bay have access to abundant food and can forage within 10 km of their colony (Regular, Hedd & Montevecchi, 2013). Time on water per day (%) may also vary with food abundance for the rhinoceros auklet. Although we did not measure time spent on water per day (%) for the same individual auklets for which we measured DEE, their energy expenditure while resting on the sea was estimated to be 261.4 kJday1, or 26.0% of the DEE if they spent the same time resting on the sea within their LCT as in the previous study (Kato, Watanuki & Naito, 2003).

Conclusions

Many studies of seabird energetics have concentrated on quantifying the energetics of flying and diving because such locomotion is considered costly (Elliott et al., 2013b). However, seabirds spend considerable amounts of their time at all seasons resting on the sea. In this study, we have shown that the RMR of resting auklets is elevated, particularly at temperatures below their LCT on water, compared with in air. Accordingly, spending time above their LCT on water provides enhanced benefits, particularly to seabirds such as rhinoceros auklets which rest for a considerable amount of time on water each day.

Supplemental Information

Supplemental Information 1 The details of body mass and resting metabolic rates in air and on water

Click here for additional data file.

Supplemental Information 2 The details of body mass, initial injectate (moles), injectate enrichment, background enrichment, initial enrichment and increments for isotope injection

Body water pool was calculated from oxygen isotope enrichments.

Click here for additional data file.

Supplemental Information 3 The details of body mass, isotope turnover rates (hydrogen and oxygen), dilution space and field metabolic rates

Click here for additional data file.

We are grateful to M Aotsuka, Y Watanuki, M Yamamoto, S Hashimoto, A Takahashi, N Sato and U Shimabukuro M We would also like to thank M. Brazil, Scientific Editing Services, for help with the preparation of the final manuscript and Dr. K Welch Jr, Dr. KElliot, Dr. JA Green and an anonymous referee for their many comments for improvements to this manuscript.

Additional Information and Declarations

Competing Interests

Author Contributions

Animal Ethics

Field Study Permissions

Data Availability

The authors declare there are no competing interests.

Aika Umeyama conceived and designed the experiments, performed the experiments, authored or reviewed drafts of the paper, and approved the final draft.

Yasuaki Niizuma conceived and designed the experiments, performed the experiments, analyzed the data, prepared figures and/or tables, authored or reviewed drafts of the paper, and approved the final draft.

Masaki Shirai analyzed the data, prepared figures and/or tables, authored or reviewed drafts of the paper, and approved the final draft.

The following information was supplied relating to ethical approvals (i.e., approving body and any reference numbers):

The Animal Experimental Committee of Meijo University approved this research (2015-A-E-5, 2016-A-E-10, 2017-A-E-2).

The following information was supplied relating to field study approvals (i.e., approving body and any reference numbers):

The Ministry of the Environment and the Agency of Cultural Affairs (21-26-0291 0292, 21-27-0367 0368 0369 0370 0371, 21-28-0344 035 036 037) and the Agency of Cultural Affairs (26-4-2188, 27-4-1928, 29-4-18) approved fieldwork.

The following information was supplied regarding data availability:

The raw measurements are available in the Supplemental Files.

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
