# Peer review of "Field and laboratory metabolism and thermoregulation in rhinoceros auklets"

_PeerJ, doi:10.7717/peerj.11460_

## Round 0.1 · original submission · Major Revisions

As you will see from the reviewers' comments, there was consensus that the data your team has gathered is welcome and worthy of presentation. However, two reviewers are not convinced that these data address the central question you pose. Therefore, a substantially re-framed presentation of your study is warranted.

Reviewer 1 ·

Basic reporting

The article is very well written. There are a few minor places where there is some syntax/grammar issues, but these can easily be edited.

The literature background is extensive and well covered.

The figure and table are appropriate. However, I do not see any reference to the raw data.

Experimental design

I am not an expert on the calculations related to RMR and DEE, I will assume, based on the supportive literature, that the analytical steps taken are the most appropriate at this time.

Validity of the findings

No raw data presented.

Additional comments

Some minor comments, mostly by line:

Line 91 – grammatical error…

Curious – were birds banded so that they were known to have participated in this experiment?

Is there any speculation on what happened to the other 7 birds that were used for the DLW field experiment?

316-317..something about the change/transition between sentences is awkward

321: Period after Elliot 2013a. ??

333: Edit the sentence: "..but common murres likely spend less time on the water as they have more frequent feeding return trips.” Or some such slight modification.

333+ This sentence is awkward: "The feeding behavior of auklets could make them to stay longer time to digest food following diving bouts (Elliott et al., 2014) and then to increase their metabolic rate for the obligatory component of the heat increment of feeding (Hawkins et al., 1997)."

The last paragraph starting on page 325 needs better formatting of the ideas. Perhaps it can be divided into two or more smaller paragraphs? Discuss thermoregulatory issues, and then the issues of timing to deliver food back to the colony (one trip per day vs multiple trips)? Maybe a new paragraph starting on line 336????

In the conclusion: perhaps the last sentence might better say something that this high energetic cost (26.6%) may be a significant energetic concern for some seabirds, especially when they are in waters below their LCT. The point is that this time, when historically has been ignored as not energetically costly (like flying or diving), may indeed, be significant.

Table 1: I would add the genus and species name to the table. I would also italicize the data from THIS study

·

Basic reporting

The paper is clearly written and presented.

Experimental design

The experimental design is suffiicent to answer the question they are asking (see direct comments below).

Validity of the findings

Underlying conclusion is robust.

Additional comments

Seabirds, especially auks, spend a lot of their life sitting on the water, but this is seldom thought to be an important component of their daily energy expenditure--although clearly an activity that takes considerable time will contribute significantly to energy budgets even if it isn't particularly costly. The authors make detailed measurements of daily energy expenditure and resting metabolic rate, and illustrating that sitting in cold water can be a significant contributor to energy budgets. These conductivity curves and daily energy expenditure measurements are technically challenging; kudos to the authors for completing them.

I have only a few minor suggestions:

L36 Suggest "on thermoregulation" rather than "for thermoregulation".

L60. Suggest adding "The" before "rhinoceros".

L. 106. Was CO2 production measured? It was be nice to have a robust estimate of RQ for DLW.

L 165. Could there be differential partitioning of isotopes into RBC water and plasma water? Usually water from whole blood is used.

L 217. The authors have validated their DLW measurements with direct respirometery. This is an important step that is seldom done, and provides great confidence in the results.

L. 313. A number of authors have tried to use similar analyses to your own to estimate energy costs year-round (Fort et al. 2009; Dunn et al. 2020). For example, Fort et al. suggest that the cold water in late winter is an energetic bottleneck and that birds switch between warm and cold water of the Golf Stream for that reason. I think it would be worth commenting on the use of your data for estimating energy costs year-round--as others are likely to do so.

Fort, J., Porter, W.P. and Grémillet, D., 2009. Thermodynamic modelling predicts energetic bottleneck for seabirds wintering in the northwest Atlantic. Journal of Experimental Biology, 212(15), pp.2483-2490.

Dunn, R.E., Wanless, S., Daunt, F., Harris, M.P. and Green, J.A., 2020. A year in the life of a north Atlantic seabird: behavioural and energetic adjustments during the annual cycle. Scientific reports, 10(1), pp.1-11.

L. 333. When capelin is present, common murres at Witless Bay have access to very abundant food, and can forage very close to the colony (~10 km, compared to ~50 km for Coats and Hornoya; Regular et al. 2013). This likely explains why they can spend so much time resting on the water and very little flying.

Regular, P.M., Hedd, A. and Montevecchi, W.A., 2013. Must marine predators always follow scaling laws? Memory guides the foraging decisions of a pursuit-diving seabird. Animal Behaviour, 86(3), pp.545-552.

·

Basic reporting

1. The background and introduction to the paper set the study up to focus on the high energetic costs of time spent on the sea surface for seabirds, species which must deal with the contrasting thermoregulatory challenges of life in the water, the air and the land. However as noted elsewhere in my review, I do not think that this is a very successful or helpful angle for this dataset. The data do not particularly support the thesis and as such this background is not very helpful to introduce what the data could instead be used to show, which is a more general exploration of the energetics of this species, which would be valuable contribution to the literature.
2. The paper is well written enough to be understood for the most part, though the use of the word ‘could’ at times is not easy to understand and it is possible that this word is being applying incorrectly. ‘Could’ would normally be used in circumstances were an inference or interpretation is not certain or speculative, whereas at times in the paper it appears that it is used to be more certain or positive that something is happening. This is distracting and some messaging is lost.
3. The word ‘may’ is also used out of context. For example in Line 43 – I think it is clear that seabirds have many adaptations for thermoregulation in water, as the paragraph goes on to describe. So the word ‘may’ is unnecessary.
4. The figure is too small to view and assess properly. The table is fine.
5. The Introduction of your paper would benefit from a much stronger final sentence or two which clearly lays out the specific aims, questions or hypotheses that your study addresses. For example this would help the reader understand why there is a section in the Methods and Results on ‘Data obtained from the literature’. As written, this comes as a surprise to the reader. As a reader, it really helps to know as early as possible what to expect and why. If there is objective in your Introduction to make an explicit comparison to existing data then I will be expecting this section later on.
6. Similarly, your Discussion would benefit by starting with a restatement of the main findings, in consideration of the aims, questions or hypotheses set at the end of your Introduction. Jumping straight into literature and allometric comparisons of RMR and LCT is not a compelling way to start this section.

Experimental design

1. In methodological terms, everything seems to have been conducted sensibly and appropriately. Respirometry and DLW techniques have been applied appropriately though some details could be clarified as listed in the following comments.
2. Line 92. Specify how many birds were captured each year and how many for in air and how many for on water measurements.
3. Line 92. Do auklets carry prey in their bills or in their stomachs? Did you consider them to be post-absorptive? How might digestive status have affected your measurements and how might this affect the in air and on water measurements differently?
4. Lines 107-109. Specify how many individuals were tested at each temperature in both air and on water. Clarify how temperatures were selected and that there were discrete temperatures used for both air and water (as suggested by Figure 1). Clarify whether each bird was only used at one temperature and only used for in air or on water (or both).
5. Lines 115-117. If you saw an appreciable mass loss overnight, did you consider using this to estimate energy expenditure?
6. Line 154. Please clarify what time of day the initial enrichment occurred for the auklets. They were recaptured at night and the Results suggest approximately a 24 hour period until recapture, but this should be clearer.
7. Lines 227-230. This is confusing. These equations suggest that there is no thermoneutrality in water in these species and that these equations would apply at any water temperature. However as noted later on, the two guillemot species do appear to have an LCT in water (Croll & McLaren 1993).
8. Lines 230/231. If units are being converted from W (which is a rate, J/s) then there needs to be a time component to the converted rate. Joules per what unit time?

Croll, D.A. & McLaren, E. (1993) Diving metabolism and thermoregulation in common and thick-billed murres. Journal of Comparative Physiology, 163, 160-166.

Validity of the findings

1. My major concern with the paper is the conclusion that time spent on water is a large component of DEE for this species. The data presented do not demonstrate this point. The auklets spend ~50% of their daily time-budget on water but only 25% of their energy budget. To me this suggests that time on water represents a considerable energy saving when compared to flying most obviously. Furthermore this ratio of time to energy is nearly identical to the two other species and three other studies presented in the paper. As such, the finding is in no way remarkable and certainly not different to the other species presented. This is not a query about novelty, but an observation that the data do not support the conclusion for this species which underlies the current thesis of the paper.
2. Another important point is the way in which the total energy expenditure of time on water during free-ranging is calculated. The animals dosed with DLW did not have their time-budget or location recorded. Thus the estimate of total energy expenditure on the water comes from an assumption both about the time spent and the temperature, both of which will influence this sum. Thus to make this the main finding when based on two estimates seems odd. Better to focus on the difference between air and water measured in the laboratory, which you can be much more sure of and compare to other seabirds. I recommend separately reporting the DEE from DLW, a value which is always welcomed for a new species and new study sytem.
3. The findings of the study seem valid in terms of the values that are produced. As noted in the manuscript, the values are consistent with similar species which gives confidence in the results.
4. Only 4 of 11 auklets injected with DLW were recovered. This is a low return rate and one of the four birds did not return for three days, which seems like quite a long time. This species is known to be prone to investigator disburbance (Sun et al. 2020). Please comment on the return rate and what implications this might have for your study findings. In particular, how representative might the behaviour of your birds be?
5. Line 275. Did you consider comparing your DEE measurement to predictions from the Seabird FMR Calculator (https://ruthedunn.shinyapps.io/seabird_fmr_calculator/) and accompanying paper (Dunn, White & Green 2018) which is the most up to date meta-analysis of seabird DEE?

Dunn, R.E., White, C.R. & Green, J.A. (2018) A model to estimate seabird field metabolic rates. Biology Letters, 14, 20180190.
Elliott, K.H., Ricklefs, R.E., Gaston, A.J., Hatch, S.A., Speakman, J.R. & Davoren, G.K. (2013) High flight costs, but low dive costs, in auks support the biomechanical hypothesis for flightlessness in penguins. Proceedings of the National Academy of Sciences, 110, 9380-9384.
Sun, A., Whelan, S., Hatch, S.A. & Elliott, K.H. (2020) Tags below three percent of body mass increase nest abandonment by rhinoceros auklets, but handling impacts decline as breeding progresses. Marine Ecology Progress Series, 643, 173-181.

Additional comments

1. At times the paper talks about the conductivity of ‘salt water’ but the on-water respirometry work was conducted in fresh water so probably safest to just talk about conductivity and time spent on ‘water’.
2. Line 81-83. This is an odd statement and seems out of place here in the final setup of the paper.
3. Line 91. Repetition of ‘in air and on water’.
4. Lines 278297. This extensive discussion of LCT in polar seabirds is interesting, but not really relevant to the goals of your paper, especially since you did not specifically determine LCT for your study animals. This could be helped by restricting the Introduction and Discussion as recommended above.
5. Lines 288-301. Similarly, this discussion is interesting, but barely relevant to your work as you did not look at metabolic costs during flight and did not fully explore LCT or UCT of your birds in your experiments. However you could have attempted to explore flight costs, using the a modified approach at the population level developed in previous studies, which would be a very valuable addition to your work (Elliott et al. 2013). As noted above, because of the low cost of time on water compared to high time investment, flight costs must be very high. Since you can estimate the costs on land and in water and estimate time in all activities from the previous study, you could also estimate the energy cost of flight and compare to published models and estimates and measurements from other alcids.
6. Lines 316-324. You seem to be suggesting that your auklets and common guillemots from Witless Bay spend more time at sea because they are in water that is inside their TNZ. This appears to be what you think is your major finding, but as noted, it is so well hidden inside the paper that it is hard to extract this. I am also not convinced of this conclusion since the temperature experienced by the Witless Bay guillemots is still less than the LCT reported for this species (Croll & McLaren 1993). I am also not at all convinced by this argument since if you divide the %time on water by %energy spent on water, the ratios are nearly identical across the four studies and the top two are certainly not better than the bottom two from Table 1. This implies that as time on water increases the proportion of energy expended on water increases at broadly the same rate, across all four studies. Indeed compared to flight, as with the other studies of auks, it could be argued that the auklets save energy by staying on the water. This is indeed consistent with a nocturnal strategy found in this species.
7. Lines 341-353. I found this element of the Discussion not that easy to follow and not well connected to the data that you collected.
8. Lines 356-361. Overall while this statement is correct, as I have attempted to explain elsewhere, this is not the most interesting or easy to reach conclusion that could be explored with these data. As noted, the amounts are not remarkable compared to other alcids and so I do not think that structuring your paper around this idea has served you or your data very well.

Croll, D.A. & McLaren, E. (1993) Diving metabolism and thermoregulation in common and thick-billed murres. Journal of Comparative Physiology, 163, 160-166.

Elliott, K.H., Ricklefs, R.E., Gaston, A.J., Hatch, S.A., Speakman, J.R. & Davoren, G.K. (2013) High flight costs, but low dive costs, in auks support the biomechanical hypothesis for flightlessness in penguins. Proceedings of the National Academy of Sciences, 110, 9380-9384.

---

## Round 0.2 · Major Revisions

As you will see, one of the reviewers feels that their major critique of the manuscript was not sufficiently addressed. Specifically, this reviewer does not agree that the framing of the manuscript around the question of the relative importance of resting on the sea surface to total energetics is not where the strength of this article lies. Instead, both reviewers note the value of these data as an exploration of temperature effects and as valuable additions to the literature. I suggest the authors focus on restructuring the presentation of these data.

In addition, several important suggestions were implemented in part of the manuscript, but not elsewhere. For example, the reviewers recommended removing "salt" from discussions of the conductivity of water vs air (since the authors employed "fresh" water for their lab study). However, while this word is now removed from the main text, it is still present in the abstract.

·

Basic reporting

The authors have addressed all of my concerns in their rebuttal.

Experimental design

The authors have addressed all of my concerns in their rebuttal.

Validity of the findings

The authors have addressed all of my concerns in their rebuttal.

Additional comments

The authors have addressed all of my concerns in their rebuttal.

·

Basic reporting

I am concerned that the authors failed to pick up my major criticism of the previous version due to this review format. Therefore I have attached a more conventional review this time.

Experimental design

I am concerned that the authors failed to pick up my major criticism of the previous version due to this review format. Therefore I have attached a more conventional review this time.

Validity of the findings

I am concerned that the authors failed to pick up my major criticism of the previous version due to this review format. Therefore I have attached a more conventional review this time.

Additional comments

I am concerned that the authors failed to pick up my major criticism of the previous version due to this review format. Therefore I have attached a more conventional review this time.

---

## Round 0.3 · Minor Revisions

As you will see. Both reviewers agree that you have adequately addressed their major concerns. One reviewer feels that the grammar can be improved and has returned an extensively annotated copy of your manuscript (which you should find attached). If you are able to address all of these remaining issues, I will be pleased to accept your manuscript for publication.

·

Basic reporting

The authors have carefully considered the comments provided by both myself and the other reviewer and altered their paper. Perhaps the main result that the proportion of the daily energy budget associated with resting is proportional to time spent resting isn't particularly surprising, but I believe these are nonetheless challenging measurements to make and are worth reporting. I have no issue with their actual measurements (and I don't believe the other reviewer had substantial concerns either; it was mostly about the interpretation of those measurements). The calculation of DEE from DLW can vary by +/- 50% depending on the equation used (see Shaffer below), and RMR in the wild for a burrow-nesting auk is likely to be substantially lower due to the absence of stress combined with metabolic suppression in the burrow. Thus, there is considerable uncertainty in what these measurements actually mean and how to compare them. Nonetheless, they're the best we have, and well worth reporting.

Shaffer, S.A., 2011. A review of seabird energetics using the doubly labeled water method. Comparative Biochemistry and Physiology Part A: Molecular & Integrative Physiology, 158(3), pp.315-322.

The one point that is still missing is to fully acknowledge the potential for bias due to capture. I suggest a line in the methods such as "Nonetheless, we acknowledge that capture may cause a stress response whereby birds spend considerable time preening on the water leading to a low DEE (Schultner et al., 2010), especially for auklets, which are known to be particularly sensitive to disturbance (Sun et al. 2020) as shown by our low recapture rate." Or something similar.

Experimental design

No comment.

Validity of the findings

No comment.

Additional comments

No comment.

·

Basic reporting

The manuscript is acceptably structured and written. In some sections, particularly new material, the standard of English is lower. I have tried to indicate these sections on the annotated pdf to help make final improvements.

Experimental design

All previous concerns have been attended to. Methods are adequately and repeatably described. The research question is now simply defined and for the most part answered. However the final message of the study could be clearer, as indicated on the annotated pdf.

Validity of the findings

The authors have restructured the paper in line with my suggestions and the findings are now valid in terms of how the measurements were made and what they might mean. I have made some suggestions on the annotated pdf for final improvements in this respect.

Additional comments

Thank you to the authors for rewriting the paper. I have made some final suggestions to help expression and aid in the readability of your work.

---

## Round 0.4 · Minor Revisions

I'm sorry for the slow turnaround of this manuscript. I'm pleased to inform you that I consider it acceptable for publication. However, as PeerJ does not provide extensive copyediting by staff, I took the time to do this myself. I did find a few small typos or grammatical errors and adjusted some formatting to make it more consistent throughout. If you simply implement the edits I have made, I will immediately "accept" that version for publication. I apologize for not sending a .doc file, but the system will only let me attach a PDF.

Thank you again for your hard work on this manuscript.

---

## Round 0.5 · accepted · Accept

Thank you. And congratulations.